# A map of global peatland distribution created using machine learning for use in terrestrial ecosystem and earth system models

Yuanqiao Wu<sup>1</sup>, Ed Chan<sup>2</sup>, Joe R. Melton<sup>3</sup>, and Diana L. Verseghy<sup>4</sup>

<sup>1</sup>Firmex Inc. 110 Spadina Avenue, Suite 700, Toronto, ON M5V 2K4, Canada

<sup>2</sup>Climate Research Division, Environment and Climate Change Canada, 4905 Dufferin Street, Toronto, ON, M3H 5T4, Canada

<sup>3</sup>Climate Research Division, Environment and Climate Change Canada, Victoria, B.C., Canada

<sup>4</sup>Retired, formerly Climate Research Division, Environment and Climate Change Canada, 4905 Dufferin Street, Toronto, ON, M3H 5T4, Canada

Correspondence to: Ed Chan (ed.chan@canada.ca), Joe R. Melton (joe.melton@canada.ca)

## Abstract.

Peatlands store large amounts of soil carbon and constitute an important component of the global carbon cycle. Accurate information on the global extent and distribution of peatlands is presently lacking but it important for earth system models (ESMs) to be able to simulate the effects of climate change on the global carbon balance. The most comprehensive peatland

- map produced to date is a qualitative presence/absence product. Here, we present a spatially continuous global map of peatland fractional coverage using the extremely randomized tree machine learning method suitable for use as a prescribed geophysical field in an ESM. Inputs to our statistical model include spatially distributed climate data, soil data and topographical slopes. Available maps of peatland fractional coverage for Canada and West Siberia were used along with a proxy for non-peatland areas to train and test the statistical model. Regions where the peatland fraction is expected to be zero were estimated from
- a map of topsoil organic carbon content below a threshold value of 13 kg/m<sup>2</sup>. The modelled coverage of peatlands yields a root mean square error of 4% and a coefficient of determination of 0.91 for the 10,978 tested 0.5 degree grid cells. We then generated a complete global peatland fractional coverage map. In comparison with earlier qualitative estimates, our global modelled peatland map is able to reproduce peatland distributions in places remote from the training areas and capture peatland hot spots in both boreal and tropical regions, as well as in the southern hemisphere. Additionally we demonstrate that our
- machine-learning method has greater skill than solely setting peatland areas based on histosols from a soil database.

The works published in this journal are distributed under the Creative Commons Attribution 3.0 License. This licence does not affect the Crown copyright work, which is re-usable under the Open Government Licence (OGL). The Creative Commons Attribution 3.0 License and the OGL are interoperable and do not conflict with, reduce or limit each other.

©Crown copyright 2017

## 5 1 Introduction

Peatlands contain 20% of the global soil carbon stocks and have played an important role in regulating the global climate since the onset of the Holocene (Yu et al., 2010). Earth system models (ESMs) simulate the global carbon cycle and its feedbacks to climate and are used to make future climate projections. Recognizing the importance of carbon stored in peatlands for the global carbon budget, climate modelling groups have begun to integrate peatlands into stand-alone terrestrial ecosystem models

- (TEMs) and also TEMs that serve ESMs and global climate models (GCMs). For example, wetlands and peatlands have been incorporated into the Lund–Potsdam–Jena (LPJ) model to simulate global methane emissions (LPJ-WHy; Wania et al. (2009b, a); Spahni et al. (2013)) and the spatial expansion and carbon sequestration of peatlands as well as wetlands (Kleinen et al., 2012; Schuldt et al., 2013) during the Holocene. A peatland carbon model has recently been developed for the Canadian Land Surface Scheme and Canadian Terrestrial Ecosystem Model (CLASS-CTEM) which forms the terrestrial component of the
- Canadian Earth System Model (CanESM) (Wu et al., 2016).

Since the land surface of GCMs and ESMs is grid-based, a prerequisite of integrating peatlands into these models involves defining the location and the fractional cover of peatlands on the model grid. However, peatlands have generally been over-looked in landscape databases and their mapping remains challenging (Krankina et al., 2008). Since peatlands are usually considered as a type of wetland that contains large amounts of organic carbon in the soil, one previous approach to determining

- peatland distribution has been based on maps of soil organic matter density (e.g. Wania et al. (2009b)). However, using soil organic matter databases alone in determining peatland distribution tends to overlook the subsurface hydrology and vegetation (more on this in Section 3.2). The first complete global peatland distribution map derived from a paleontological perspective was produced in 2010 (Yu et al., 2010), but it is an estimated binary map, not a gridded product, and it does not provide quantitative information on fractional coverage. As stated by Yu et al. (2010) in describing their dataset, "accurate true peatland
- coverage and distribution is not available for many mapped regions".

Another approach has been to use a soil map together with global wetland maps or inundation extent maps (e.g. Köchy et al. (2015)). Wetland and inundated area databases have mostly been produced using the following techniques: mapping of shallow surface water based on remote sensing data as in the Global Inundation Extent from Multi-Satellites (GIEMS) initiative (Prigent et al., 2007; Papa et al., 2010) and the Surface WAter Microwave Product Series (SWAMPS) (Schroeder et al., 2015); and land

cover mapping using surface observations and moderate resolution imaging spectroradiometer (MODIS) data as in the Global Lake and Wetlands Database (GLWD-3; Lehner and Döll (2004)). However, the currently available wetland mapping products are of limited utility for peatland modelling applications. These databases generally do not agree well amongst themselves (Melton et al., 2013) and may exhibit biases depending on how they were generated (see discussion in Bohn et al. (2015)). As

well, in the boreal zone some peatlands are not inundated, therefore using hydrological characteristics alone can underestimate their extent (Matthews, 1989; Prigent et al., 2007).

The reliability of maps based on remote sensing data depends on the sampling and interpretation of the remote sensing signals and the quality control methods (e.g. Zhao et al. (2014)), and is complicated by technical issues such as the interference of trees with the signals (Krankina et al., 2008). Because of the difficulties with ground truthing, good quality peatland

- ence of trees with the signals (Krankina et al., 2008). Because of the difficulties with ground truthing, good quality peatland coverage maps are only available for limited areas such as Canada (Tarnocai et al., 2011), Sweden (SGU, 2011) and West Siberia (Peregon et al., 2009). Despite recent advances in remote sensing, mapping peatlands globally at a high resolution also requires knowledge of pedology and palaeoecology, and a complete map using these methods is not expected until at least 2020 (Barthelmes et al., 2014).
- We are aware of two attempts to dynamically determine peatland extent on a global scale. In their 2012 paper, Kleinen and coworkers used a TOPMODEL-based approach to estimate wetland area and water table depth (Kleinen et al., 2012). A statistical approach based on topography was used to determine regions that are likely to flood. Regions with relatively stable wetland areas were then allowed to eventually grow peat. Stocker et al. (2014) adapted the TOPMODEL-based approach of Kleinen et al. (2012) and brought it to the global scale in their Dynamical Peatland Model based on TOPMODEL (DYPTOP).
- Both models are designed to simulate peatlands as well as other forms of wetlands. An additional study by Thompson et al. (2016) used forest inventory plots along with organic soil depth measurements to create a predictive model of treed peatlands across Canada. Their model found that certain tree species, along with stand height and stand age, were the best predictors of peatland presence. Unfortunately, detailed information like this is not available on a global scale. We present here an approach based on machine learning techniques that does not rely upon the TOPMODEL formulation or forest inventory plots, and that
- can independently determine peatland locations globally.

#### 2 Materials and Methods

## 2.1 Data acquisition and preparation

The general process of data preparation, model training and testing is illustrated in Fig. 1. To account for the hydrological and climatic criteria for peatland formation, we used input data covering three aspects of the land surface: climate, soil and topography. Climate data were obtained from the comprehensive Climatic Research Unit (CRU) database, version 3.22 (University of East Anglia Climatic Research Unit et al., 2008). The monthly mean and annual mean values of cloud cover, diurnal temperature range, potential evapotranspiration, precipitation, minimum daily temperature, mean daily temperature, maximum daily temperature, and vapour pressure were generated from the 1901 to 2013 monthly time series. In total, therefore, 104 climatic input variables for the statistical model were produced from the 8 climate variables listed above (12 monthly average

values and one annual average value per climate variable). Soil properties were obtained from the Harmonized World Soil Database (HWSD) v1.2 (Wieder et al., 2014) at 0.05 x 0.05 degree resolution, which was regridded and interpolated to 0.5 x 0.5 degree resolution. Twenty-three soil properties were obtained, including: available water storage capacity, sum of subsoil C content, sum of topsoil C content, soil or non-soil units, depth of obstacles to roots, topsoil bulk density, subsoil bulk density,

topsoil gravel content, subsoil gravel content, topsoil clay fraction, subsoil clay fraction, topsoil organic carbon, subsoil organic carbon, subsoil organic carbon, topsoil sand fraction, subsoil sand fraction, topsoil silt fraction, subsoil silt fraction, topsoil pH in water, subsoil pH in water, cation exchange capacity of the clay fraction in the topsoil, cation exchange capacity of the clay fraction in the subsoil, cation exchange capacity of the clay fraction was incorpo-

- rated by using calculated fractions of each 0.5 x 0.5 degree grid cell with slopes below specified thresholds using the digital elevation ETOPO1 data (Amante and Eakins, 2009). The ETOPO1 data were used to calculate slopes at 1 arc minute (1/60th degree) resolution. Each 1 arc minute grid cell was assigned a slope that was the average of eight slopes based on its elevation and the elevation of its eight surrounding grid cells without consideration of aspect. The fraction of each 0.5 degree grid cell that was flatter than a given slope threshold was calculated using these 1 arc minute slopes. Eight slope thresholds were used:
- 0.35%, 0.30%, 0.25%, 0.20%, 0.15%, 0.10%, 0.05% and 0.025%. The total number of input variables for the statistical model was therefore 135.

For training and testing the model, peatland fractional cover was selected as the target variable. Peatland coverage data were obtained for Canada (Tarnocai et al., 2011) and West Siberia (Peregon et al., 2009), where 12% and 50-75% respectively of the land surface is covered with peatlands. For Canada, ESRI shapefiles were available with information on Bog, Fen and Bog/Fen

- features with ≥1% peat coverage (Tarnocai et al., 2011). Fractional peatland cover was projected from these polygons onto the 0.5 x 0.5 degree grid. The West Siberia dataset contains peatland fractional cover on a 0.5 x 0.5 degree grid aggregated from remote sensing and ground survey based data describing 20 wetland types and their areal cover (Peregon et al., 2009). Since it is unlikely that a reliable global map of peatland coverage can be generated using data solely from these two regions in the Northern Hemisphere, we derived additional data to provide coverage outside of Canada and West Siberia. Areas where there
- are no peatlands at all should correspond to sufficiently low amounts of soil organic carbon. We set the peatland coverage to zero for all grid cells below a threshold topsoil organic carbon content (from the HWSD dataset) of 13 kg/m<sup>2</sup>, which was the value that provided the best fit during the training and testing of the model. We experimented with other variables that could potentially be used as a proxy, such as subsoil organic carbon content and annual precipitation, but none of these produced a better fit.

#### 25 2.2 Extremely randomized tree methods

In light of recent successful applications of machine learning methods to global mapping in various areas (e.g. Crowther et al. (2015)), we set out to produce a spatially continuous global peatland fractional coverage map, using machine learning and available information on peatland distribution, which would be suitable for use as an input geophysical field for TEMs/ESMs. Extremely randomized trees, or Extra-Trees, is an ensemble, nonparametric tree model for data interpretation and statistical

modelling. A classification or a regression tree is a representation of an input-output model by a tree whose interior nodes are each labelled with a test based on one input variable (Geurts and Louppe, 2011). Each terminal node of the tree is labelled with a value of the output. The predicted output of the target variable is determined as the output associated to the leaf propagating through the tree starting at the root node. A tree is built by recursively identifying at each node the test that leads to a split of the node sample into two subsamples that are as pure as possible in terms of their output values (Geurts and Louppe, 2011).

Extra-Trees uses randomly selected features at each node in each tree, but randomizes strongly both the attribute and cut-point choice when splitting a tree node (Geurts et al., 2006), which significantly improves precision and reduces computational complexity while increasing computational efficiency and scalability (Wehenkel et al., 2006). Extra-Trees is especially suited to *batch-mode* supervised learning problems with a focus on those characterized by numerous input variables and a single

5 target variable (Geurts et al., 2006). Therefore, it is suitable for this study on peatland mapping where numerous variables that may be correlated to varying degrees are being used as predictors. To select the most informative features from the original 135 identified, we used the L1-based cross validation feature selection tool that is implemented in the scikit-learn library. L1 prior, or Lasso, is a linear model that estimates sparse coefficients, which is usually used for sparse estimators due to its tendency to prefer solutions with fewer parameter values (Pedregosa et al., 2011).

#### 10 2.3 Statistical modelling and evaluation

The statistical modelling was conducted in the programming language Python 2.7.10. The Scikit-Learn library (Pedregosa et al., 2011) was used for building, evaluating and optimizing the Extra-Trees model. As noted above, 135 input variables were prepared and parsed into the model. Using the default LassoCV hypo-parameters in scikit-learn, however, only 14 of the original 135 features were selected for the final model and are listed in Table 1. It should be noted that none of the slope

thresholds was selected indicating that slope was not found to be a constraint on peatland location. The combined datasets were randomly split across the 27,443 grid cells for which we have values for peatland coverage (60% for training and 40% for testing). The model was optimized for the highest coefficient of determination ( $r^2$ ).

As fractional peatland coverage data do not currently exist for evaluating the model at a global scale, we used the qualitative peatland distribution map of Yu et al. (2010) as an independent check of our results. We projected this map, which consists of an

- irregular grid containing a logical field indicating the presence or absence of peatlands, onto a 1/24<sup>th</sup> degree latitude-longitude grid. This high resolution logical map was further interpolated using a box-averaging method onto our statistical model grid at 0.5 x 0.5 degree resolution. Peatland coverage derived in this manner depends on the assumption that the density of the points on the original grid is a proxy of the fractional coverage. Comparing this result over the two regions where we have good quantitative information on peatland coverage, the method appears to work well over West Siberia, but is problematic
- over Canada because of the lack of points in the original grid over the Mackenzie valley and in the area south of Hudson Bay (see Figure A1). In order to obtain the best global representation of peatland coverage, therefore, we merged this global map with the Canadian and the West Siberia peatland coverage maps to provide a basis for comparison against "observations". We refer to this product hereafter as C-WS-Y.

## 3 Results and Discussion

## 3.1 Peatland distributions globally and regionally

The most important variables for determining peatland locations, as found by the statistical model, include top soil organic C content (% weight), subsoil organic C content (% weight) and area weighted subsoil C content (kg C m<sup>-2</sup>). Together these
three variables explain 74.4% of the variance as found by the statistical model. Of the remaining 25.6% of the variance, climatic variables explain 22.2% (including monthly mean cloud percentage cover in November (3.9%), annual mean of the monthly near-surface minimum temperatures (3.8%), cloud percentage cover (2.6%), vapour pressure (2.6%), precipitation (1.8%), monthly mean precipitation in August (2.0%) and April (1.8%))(Table 1). These predictor variables indicate that the soil organic C content is the best indicator of peatland location. Perhaps surprisingly, other climate indices that have been suggested

- as helpful in predicting boreal peatland locations (e.g. Alexandrov et al. (2016)) such as precipitation, temperature, potential evapotranspiration, or cloud cover were all found to play a small role in the statistical model's prediction of a peatland location. The result suggested that for near-surface temperature, the average minimum is perhaps more important than the average mean temperature; for precipitation, certain months such as August and April are more critical than the other months in determining the location of peatlands. The selection of August and April could represent an artifact of the datasets available for model
- training, which are from the northern hemisphere. It is possible that the lack of training datasets for the southern hemisphere would bias the feature selection towards variables of importance to, primarily, northern hemisphere peatlands. Interestingly, slope was not selected as one of the final 14 variables. This demonstrates that prior selection of deterministic variables, as has been done in other studies (e.g. Gallego-Sala and Prentice (2012)), can lead to the use of variables that actually do not have any predictive power for determining peatland distribution.
- Our prediction error was calculated as the difference between the modelled and the observed values for the grid cells in our test regions of Canada and West Siberia. Figure 2 shows the heat map of prediction error with the training data masked in light grey. The total modelled peatland area in Canada was 1.11 million km<sup>2</sup>, compared with 1.14 million km<sup>2</sup> estimated by Tarnocai et al. (2011). The total area of peatlands in West Siberia calculated by the model was 0.70 million km<sup>2</sup>, which agrees very well with the estimated 0.69 million km<sup>2</sup> based on remote sensing and ground surveys (Peregon et al., 2009). For
- the 10,978 test grid cells, the root mean square error was 4% fractional coverage with a coefficient of determination of 0.91. Fig. 3 shows a scatter plot of the modelled versus observed peatland fractions with the biases color-coded, and it can be seen that there is little or no systematic error, which confirms the ability of the statistical model to capture the peatland distribution across regions.

This statistical model was then used to create a complete map of global peatland fractional coverage at 0.5 degrees resolution, shown in Fig. 4a. The merged C-WS-Y map described in the previous section is shown in Fig. 4b and is plotted with two different colour keys: the observation-based peatland fractions over Canada and West Siberia are plotted using the same colour palette as for Fig. 4a, but the fractions obtained qualitatively from Yu et al. (2010) are plotted with a different (red-based) colour palette. Compared to C-WS-Y, the model predicts similar patterns of peatland density in the northern hemisphere, the tropical areas and the South, with similar areas of high peatland coverage in the Hudson Bay Lowlands, the West Siberia Lowlands,

Finland, and tropical islands in Asia. It should be noted that in regions where the C-WS-Y map is solely derived from the Yu et al. (2010) dataset, it likely represents an underestimate of peatland area as the threshold for inclusion is >5% coverage, so many areas with peatlands below that threshold will be missed. The model also predicts some peatland areas that have been discovered in recent years or which lie outside of the data sources of Yu et al. (2010), for example southern Patagonia in Chile between  $52^{\circ}11$ 'S  $70^{\circ}57$ 'W and  $53^{\circ}45$ 'S  $72^{\circ}56$ 'W (Schmidt et al. 2010) and the Changuinola peat dome in Panama (I awson

between 52°11'S, 70°57'W and 53°45'S, 72°56'W (Schmidt et al., 2010), and the Changuinola peat dome in Panama (Lawson et al., 2015). (Note that according to Z. Yu (personal communication), polygons over Sweden and Tasmania were omitted from their original shapefile due to copyright reasons.)

Globally, we estimate the total area of peatlands at 4.42 million  $km^2$ , of which 3.90 million  $km^2$  is located north of 30°N. Our estimate for this northern region compares well with the value of 4.0 million  $km^2$  reported by Yu et al. (2010) and with

- the estimate of between 3.88 and 4.09 million km<sup>2</sup> by Maltby and Immirzi (1993). It is also in line with the Tarnocai et al. (2009) estimate for boreal permafrost regions of 3.6 million km<sup>2</sup>. Over the tropics (30°S to 30°N), our model produces a value of 0.43 million km<sup>2</sup>, which lies between those of 0.37 million km<sup>2</sup> reported in Yu et al. (2010) and 0.44 million km<sup>2</sup> in Page et al. (2011) but well below the recent model-based estimate of 1.7 million km<sup>2</sup> by Gumbricht et al. (2017). The very large extent estimated by Gumbricht et al. (2017) requires extensive ground validation, as the authors themselves state, before it can
- be assumed to be more reliable than the lower estimates of Page et al. (2011) and Yu et al. (2010). For the region south of  $30^{\circ}$ S, our estimate is 0.07 million km<sup>2</sup>, which is higher than the value of 0.045 million km<sup>2</sup> by Yu et al. (2010). Over Southeast Asia, our estimate of 0.17 million km<sup>2</sup> is lower than that reported by Page et al. (2011) of 0.24 million km<sup>2</sup>.

Regional peatland maps derived from the model and from C-WS-Y for Russia and Indonesia are shown in Fig. 5. The model predicts hotspots of peatlands with more than 40% of areal coverage along the shore of the White Sea and in the Amur River

- watershed in Russia (Fig. 5a) and in areas along the coastlines in Indonesia (Fig. 5b). The model predictions for Russia (2.17 million km<sup>2</sup>) appear to agree well spatially with C-WS-Y but are lower than Vompersky et al. (2011) (3.69 million km<sup>2</sup>). The estimate of Vompersky et al. (2011) includes both peatlands (defined by a peat layer >30 cm) and paludified shallow peatlands (defined by a peat layer <30 cm). In Indonesia, the model predicts similar locations but larger coverage of peatlands than C-WS-Y (Fig. 5b). The recent rapid loss of peatland areas due to mainly human activities may have altered the natural distribution</p>
- of the peatlands in these regions (Margono et al., 2014) and contributed to the discrepancy between the model predictions and C-WS-Y.

#### 3.2 Comparison against using the HWSD soil database for peatland distribution

Since the majority of the variance found in our machine-learning method is attributable to soil carbon content, an important test is to ensure that we have greater skill in determining peatland distribution than simply using a soil map alone. The

30 HWSD dataset includes a grid of mapping unit identifiers at 1/120<sup>th</sup> degree resolution and a database associating each mapping unit with various soil properties and characteristics, including soil types and fractional coverages. A map of histosols was constructed by assigning each mapping unit on the high resolution grid with the total coverage from soils identified as histosols according to the FAO-74 and/or the FAO-90 soil classification and box-averaged onto our 0.5 degree resolution model grid.Figure 6 shows scatter plots for Canada and West Siberia for our approach (upper panels) and the HWSD dataset (lower

panels) compared to observations (Tarnocai et al., 2011; Peregon et al., 2009). For West Siberia, the modelled peatland distribution has excellent agreement with the observations ( $r^2$  of 0.97 and RMSE of 0.005) while the HWSD shows a bias at higher peatland extent and larger error ( $r^2$  of 0.85 and RMSE of 0.012). The modelled peatland distribution for Canada shows similarly good agreement with observations ( $r^2$  of 0.95 and RMSE of 0.007) while the HWSD appears to have some threshold behaviour around 0.7 and 0.2 for peatland extent, causing greater error in peatland distribution ( $r^2$  of 0.67 and RMSE of 0.021).

5 behaviour around 0.7 and 0.2 for peatland extent, causing greater error in peatland distribution ( $r^2$  of 0.67 and RMSE of 0.021). These plots demonstrate that our method produces a significantly better estimation of peatland distribution than solely using the HWSD dataset.

## 3.3 Weaknesses of our machine-learning approach

The purpose of our study is to produce a map of peatland distribution for use as an input geophysical field for TEM/ESMs with 10 integrated peatland models. It is tempting to ask whether our technique can give any insights into peat formation or the conditions necessary for a peatland to develop and persist. Given that the machine-learning approach is using presently observed conditions to determine peatland area extent, it is difficult to determine cause from effect. While the climatic conditions that explained about a quarter of the variance are likely indicative of conditions needed for peatland formation or persistence, the majority of the variance is explained by soil carbon content. Is high soil C content required for peatland existence or is it that

15 peatlands themselves create high soil C contents? Our machine-learning method is unable to answer this rather basic question. Although useful for determining peatland location, which is our primary goal, it is unable to deal with questions regarding peatland processes at a mechanistic level.

An additional constraint of the machine-learning approach is the reliance on representative inputs. Recently discovered tropical peatlands in the Congo Basin (Dargie et al., 2017) and Amazonia (Draper et al., 2014) are not well captured by our

- 20 method as both regions do not show elevated soil carbon contents (which is the source of the majority of the variance in our method) in the HWSD soil maps. Thus this shortcoming of the HWSD dataset is passed on to our results. Future improvements in soil mapping products should yield improvements in future versions of our peatland distribution map but because our technique is diagnostic, not predictive, we will remain constrained by the quality of our input datasets. This constraint is not limited to soil maps but also to the quality of the climate and topographic datasets.
- A final weakness of our approach lies in the availability of training data. Our training data for peatland distribution is biased towards the northern hemisphere. While we have good coverage of peatland presence in Canada and West Siberia and peatland absence globally (see Section 2.1), we presently lack access to sufficient observed peatland distribution maps for the southern hemispere and the tropics for training of peatland presence in those regions. While it appears from our results that we are not heavily biased outside of the northern hemisphere (see Section 3.1), better training data would likely improve our model estimates.

## 4 Conclusions

We present a new global peatland fractional coverage map at a scale of 0.5 degree resolution. We applied a machine learning method to produce a statistical model, which was trained using existing fractional coverage datasets for Canada and West Siberia, as well as datasets of climate, soil and topographic information. Our model was able to reproduce test areas of peatland coverage within the Canadian and West Siberian regions with an  $r^2$  value of 0.91 and a RMSE of 4%. A strength of this

- coverage within the Canadian and West Siberian regions with an r<sup>2</sup> value of 0.91 and a RMSE of 4%. A strength of this peatland mapping technique is that it does not rely on wetland maps, the TOPMODEL approach, or on prescribed rules of soil organic matter density, as other studies have done. The global peatland map generated by the model successfully reproduces well-known peatland hotspots in the boreal region, tropical Asia and the Southern Hemisphere and outperforms techniques that identify peatlands solely using maps of soil classification. While our peatland map does miss recently discovered tropical
- peatlands, we attribute this to deficiencies in the datasets used as inputs to our machine-learning technique. Our global peatland map compares well against other global and regional estimates as well as against a qualitative global map (Yu et al., 2010) and is suitable for use as a peatland mask in global-scale peatland simulations both offline and as part of ESMs/GCMs, and is slated for implementation in future versions of CanESM.

## 5 Code availability

Python code for the data processing, statistical modelling and model evaluation is available on request and upon agreeing to Environment and Climate Change Canada's licensing agreement which can be viewed at http://collaboration.cmc.ec.gc.ca/science/rpn.comm/license.html. Please contact Ed Chan (ed.chan@canada.ca) to obtain the model code.

The map of peatland distribution is available at ftp://ccrp.tor.ec.gc.ca/pub/EChan/global-peatland-fractional-coverage.nc

Author contributions. Y.W. initiated the project, selected and setup the model, and wrote much of the manuscript. E. C. gridded the peatland observations shapefiles, optimized and ran the model, and generated output plots. J. R. M. advised with experimental design and setup including model testing and wrote parts of the manuscript. D. L. V. advised on the model setup and testing process and assisted in writing the manuscript. All authors contributed to the final manuscript.

The authors declare that they have no conflict of interest.

Acknowledgements. Y. Wu was supported by a Natural Sciences and Engineering Research Council of Canada (NSERC) Postdoctoral Visiting Fellowship. We are grateful to Zicheng Yu and Anna Peregon for sharing the original data of the peatland distribution maps and for valuable advice. We thank Vivek Arora and Paul Bartlett for insightful comments on the model design and on the manuscript. We also thank Thomas Kleinen for valuable input on a previous version of the manuscript.