# Peer review of "A map of global peatland distribution created using machine learning for use in terrestrial ecosystem and earth system models"

_Geoscientific Model Development, 2017_

## Short Comment (SC1) · 18 Jul 2017

Dear authors,

In my role as Executive editor of GMD, I would like to bring to your attention our Editorial version 1.1:

http://www.geosci-model-dev.net/8/3487/2015/gmd-8-3487-2015.html

This highlights some requirements of papers published in GMD, which is also available on the GMD website in the 'Manuscript Types' section:

http://www.geoscientific-model-development.net/submission/manuscript_types.html

[Figure]

In particular, please note that for your paper, the following requirements have not been met in the Discussions paper:

- "Papers describing data sets designed for the support and evaluation of model simulations are within scope. These data sets may be syntheses of data which have been published elsewhere. The data sets must also be made available, and any code used to create the syntheses should also be made available." (Editorial v1.1, Appendix A5)
  For these papers the same criteria as for model description papers apply, i.e., "The main paper must give the model name and version number (or other unique identifier) in the title." (Editorial v1.1, Appendix A2)

  In this case the "model" is the "data set".

Please add the data sets name and version number in the title in your revised submission to GMD. Additionally, as the data set seems not to be subject to licence issues. Please upload it to a perment archive providing a DOI.

Yours,

Astrid Kerkweg

———————————————————

---

## Short Comment (SC2) · 1 Sep 2017

This is a very needed and relevant work. Good globally harmonized peatland maps do not exist, and machine learning as a tool to derive these maps with better resolution and quantiative information is a track that should be gone (next to other efforts). At best, these different efforts give us soon good global peatland maps that even include information about the peatland type.

I want to limit my non-referee comment to two major points that need to be considered before this work should be accepted.

[Figure]

1) Splitting the dataset randomly (!) into training and validation must not be done when data points are correlated. In this application, there is a strong spatial correlation between the data points, i.e. pixels. The random selection of points for training means that validation is done with a highly correlated set of data points. This has two consequences: a) performance of the trained model is highly overestimated when looking at the "validation" metrics (R, RMSE, etc.) b) the training person tends to overtune/overfit the machine learning model (encouraged by the good validation) and includes more and more degress of freedom. Normally, a validation (that should be done with independent data) indicates the model developer where to stop. Recommendation: Divide your data set (left side of Fig.1) into e.g. 10 regional pieces for which you can say that they are spatially not (or only very limitted) correlated and then either use 40 % of these regions as pure validation data or perform a cross validation always leaving out only one region for which you make a prediction with a model trained on the other 9 regions. Recent publications that point out the importance of the independance of the validation data in machine learning applications: Jorda, H.; Bechtold, M.; Jarvis, N. and Koestel, J. (2015): Using boosted regression trees to explore key factors controlling saturated and near-saturated hydraulic conductivity, European Journal of Soil Science, 66(4), 744-756. Bechtold, M.; Tiemeyer, B.; Laggner, A.; Leppelt, T.; Frahm, E. and Belting, S. (2014): Large-scale regionalization of water table depth in peatlands optimized for greenhouse gas emission upscaling, Hydrology and Earth System Sciences, 18, 3319-3339.

2) Subsection 3.2: Comparison against using the HWSD soil database for peatland distribution This is not a proof that the trained model is superior than HWSD. You train a model on Tarnocai and Peregon maps while using among others HWSD as input. Then of course the trained model performs better in predicting Tarnocai and Peregon maps than HWSD. The comparison is not fair and misleading.

I hope this comment helps to improve manuscript and reliability of the map, and I hope it is taken in the spirit intended: scientific openness and fair reviewing. Supporting a

policy of open reviews and comments, I wish my name to be revealed to the authors.

Michel Bechtold KU Leuven, Belgium, 1 Sep 2017
* * *

---

## Referee Comment (RC1) · Anonymous Referee #1 · 9 Sep 2017

The authors use available soil data, climate data, and topographic data, combined with a machine learning method to produce a spatially continuous global peatland fraction map. Such a map is relevant and useful for modelling communities. They start by including all available variables and the algorithm selects the variables that have the strongest predictive power for mapping peatlands. In the end, the only variables with a lot of predictive power are soil carbon quantities. This is of course expected, given that a peatland generally has very high soil carbon - more than any other soil type - making it a good proxy.

An advantage of this method is that it does not make any assumptions about the statis-

tical distribution of the data or any of their relationships. It is useful to show that using additional variables (e.g. meteorological data) cannot give a much better distribution of peatlands than just using soil carbon alone. The method could also be very useful for other, similar applications.

The paper is very clearly written and presented, and a potentially valuable contribution, but I would recommend some substantial changes to ensure that its value is fully realised.

Comments

I wouldn't take the result that carbon is the only useful proxy as definitive. The peatland maps themselves (i.e. used for training and validation) also have relatively large uncertainties, and so the fine details of peatland distribution beyond broad classifications are probably not resolved (thus any impacts of hydrology, topography etc, may simply not be well resolved in the training dataset). However, I would also suggest trying the fraction of grid cell with topographic index higher than a certain value as an input variable, rather than basing it only on slope, because this additionally takes into account the amount of water draining into the land (essentially whether it is an upland or a lowland), which has a major impact on hydrology and the potential for peatland formation.

As in the short comment that was posted already, I do not find the evaluation of the new map very convincing, and given the uncertainties in all of the soil datasets, the extremely high accuracy that the authors claim does suggest overfitting. With using a relatively new approach in this field, I think it would be wise to go into a more thorough evaluation, perhaps discussing in more detail the form of the relationships that are produced by the algorithm, and considering whether it would be possible, for example, that using both subsoil and topsoil organic carbon content can give more information than just the fraction of histosols in the grid cell. Currently it does appear that you are using the same dataset that was input to the model, as evaluation (in Section 3.2). In

Section 3.2 there is some discussion about threshold behaviour, but could this also be a problem in the HWSD inputs to the model? And if so, if this do not show up in the modelled results, that might suggest that they have been smoothed away by overfitting to other variables. I would recommend taking the suggestions from the 'short comment' in terms of dividing the data into training and validation points, and also expanding the evaluation/discussion to make it clearer why your approach is really an improvement on simply using the histosol map.

Whether or not the approach represents a major improvement on the HWSD histosol map, it could also be applicable to other problems. The statistical modelling method is not described in much detail in the paper, however. Since this might be a direction that other modelling groups want to take, I suggest that the authors expand the description of the method, highlighting its scientific importance for Earth System Science and why it is in theory better than alternative methods. So, at the end of the introduction I would recommend adding another paragraph highlighting the basic principles of machine learning and its potential as a method for Earth System Science. Then in the methods, go into more detail in Section 2.2 (for example, what is a sparse coefficient?). References are provided, but in my opinion we should be as clear as possible in papers, particularly when these approaches are not yet well known in Earth system science (and for this journal in particular).

Comparing to Alexandrov et al. (2016), the authors say that it's surprising that climate variables don't play more of a role in their model: this is not a valid comparison. Alexandrov et al. were using climate variables alone to determine suitable conditions for peatlands, whereas the model in the present manuscript has soil carbon content as an input, which is basically a proxy for observed peatlands. So, using something close to 'observed peatlands' will of course give a closer match to observed peatlands than climate variables alone. (However, in the last glacial maximum as considered by Alexandrov et al., there is no direct observation of peatlands or soil carbon, hence the need for a predictive model based on climate and topography.)

Technical comment

On Figure 5 it's a bit difficult to see which part of the world is being shown. Could you add some latitude/longitude markers?

---

## Referee Comment (RC2) · Anonymous Referee #2 · 1 Nov 2017

Recognizing the importance of peatlands in global carbo cycling, and a lack of dedicated peatland extent maps, the authors present an attempt at broad-scale mapping of global peatland extent for use in terrestrial ecosystem and earth system models. I applaud the effort and agree that it is very important work. I also believe that the approach of combining relevant datasets in a machine learning framework is a good one. However, I see numerous problems with the method and choices of input, training and validation data. These are summarized as: (1) lack of definition of peatlands, (2) problematic choice of input variables, (3) inconsistent and auto-correlated training data and (4) the model is validated against data which is strongly correlated to the input data. See more details on the four points below. Given these issues it is impossible to

assess if the derived map has any advantages over e.g. using existing global soil maps (e.g. HWSD, WISE30sec or SoilGrids) to parameterize the extent of organic soils in models.

Given these strong limitations in the basic methods, I cannot recommend this work for publication.

1. One of the problems with harmonizing or reconciling different approaches to peatland extent mapping (or modelling) is that different definitions of what constitutes a peatland exist. My first concern is that the authors do not themselves provide a clear definition of how they have defined peatland in their study. This is problematic since your input data is based on three different definitions (see below). I recommend the authors look at e.g. Joosten and Clarke (2002, "Wise use of mires and peatlands") as a first guide in their choice. You must choose one definition that you find useful and then design your study based on that. Since your stated purpose is to improve broad-scale modelling of the carbon cycle, a definition that reflects the depth of peat accumulation seems sensible (it is also the most commonly used).

2. I also object to the choice of using the HWSD soil carbon maps as input data. Those variables in the HWSD are calculated from HWSD soil coverage. In essence, organic C% in the HWSD is so autocorrelated to peatland extent in the HWSD that you are effectively used (a derived variable of) a coarse peatland map as "independent environmental" data with training data from other peatland maps (in some cases the very same map) and finally, ground-truthing it against that same map again (because Yu et al., 2010 is largely based on the HWSD, see further below in point 4). Your model is completely dominated by HWSD organic soil carbon variables, with scattered influence of climatic variables (none of which exceed 4% explanatory power). Your model shows no strong response to topography or any of the climatic variables that are believed to influence peatland extent (temperature, precipitation etc.). Most likely, those strong signals are masked out under the driving force of the HWSD data, which implicitly already includes that information.

3. You train the model with regional peatland presence and global peatland absence. This is inconsistent and somewhat problematic. The three different datasets (a, b c below) all use different definitions of peatland. And they are mapped at very different spatial scales and have very different thematic and spatial accuracy. The regional peatland presence is from (a) thematic soil maps in Canada (the Canadian system for peatland classes is the same as the soil classification system) and from (b) thematic wetland maps in west Siberia. The latter map actually also includes non-peat forming wetland systems: quote from Peregon et al 2009: "One reason [for the higher coverage compared to earlier studies] is that our estimate comprises not only open peat-accumulating wetlands but also forested and grass-dominated wetlands with/or without peat deposits." The Global peatland absence is from (c) a threshold of topsoil organic carbon content in the HWSD. The choice of a topsoil carbon threshold is difficult to justify; carbon stocks in the HWSD are calculated based on mapped coverage of different soil types, including maps of Histosols/peatlands. How can the authors justify that the derived HWSD variable of topsoil soil carbon should be superior for peatland mapping purposes compared to the actual mapped peatland extent in that dataset? Note that this threshold presumably misses many areas where there is >13 kg C but still no peatlands (eg upland Tundra soils or Boreal soils frequently have more than 13 kg C in the topsoil without wetland or peatland conditions).

4. And as mentioned above, your environmental data, training data and validation data are all strongly correlated. Given this fact, it is of course entirely unsurprising that you model performs very well. Further the split of the dataset in random pixels for training and validation also gives a strong autocorrelation which boosts the performance metrics. This is already well covered in a comment by M. Bechtold. Note that the peatland distribution map of Yu et al. (2010) is mostly based on soil maps, and shows peatlands to exist in cases where >5% of the terrain is peat. This percentage is a rather low threshold, and may have led you to overestimate peat cover. Further the use of the map by Yu et al for assessing your model is deeply problematic, since it is in no way independent from your input training data. If you look into the supplemental materials of

Yu et al (2010) you will find that the source for peatland maps of Canada is Tarnocai et al., (2002) and for West Siberia Sheng et al., (2004). These datasets are very closely related (or near identical) to the input data used for Tarnocai et al (2011) and the base peat type map used by Peregon et al (2009). Much of the world's other peatlands in Yu et al 2010 are mapped from the HWSD Histosol (and some Gleysol) coverage. So, your validation data is almost the same data as your environmental variables and your training data.

---

## Author Comment (AC1) · 7 Dec 2017

**Reply to reviewers and note to editor**

Yuanqiao Wu, Ed Chan, Joe Melton, and Diana Verseghy

Dec 8th 2017

We thank the reviewers, Michel Bechtold, and the editor for their efforts for our manuscript. Since the submission of our manuscript another paper has been published that we believe makes our work no longer needed. The publication of Xu et al. (2018) details their PEATMAP product which is a global peatland map based primarily on mapping products and other geospatial information at global, regional, and local scales. This paper effectively provides the needed mask of peatland locations that we were hoping to achieve using our machine learning approach. As a result, in the hopes of not confusing the literature, we choose to withdraw our paper and instead draw readers' attention to the Xu et al. paper.

While we won't reply in detail to our reviewers as we are withdrawing our paper (although we do, again, wish to express our appreciation for their time in reviewing our paper), we would like to clarify one aspect that seems to be a prevalent point of confusion. If we are interpreting their comments correctly, spatial autocorrelation was raised by Michel Bechtold and Anonymous Reviewer #2. A high degree of spatial autocorrelation of the peatland locations is to be expected and is not problematic. The view that spatial autocorrelation is problematic is not uncommon but is incorrect. Kühn and Dormann (2012) have written a nice discussion on why spatial autocorrelation is sometimes assumed to be problematic while the real issue is spatial autocorrelation of residuals.

In order to investigate the degree to which the model residuals are spatially autocorrelated, we generated a spatial correlogram (Figure 1 below) using the R package "ncf" (Bjornstad, 2016). The package computes the statistic, "Moran's I" (Cliff and Ord, 1981), which takes the form of a classic correlation coefficient with values ranging from -1 to 1 (strongly negatively to strongly positively correlated) as a function of distance. Moran's I is calculated based on all pairs of values within a distance interval (lag). Distance bins of 100 km (~ 2 grid squares) were chosen for the figure, where Moran's I is plotted against the mean distance between all points within each bin. The correlogram for the model residuals is shown in red, and for comparison, the correlogram for our peatland map is shown in blue. This indicates that the peatland map is highly autocorrelated spatially (as expected) but the residuals are not with the exception of the first distance bin.

References:

[Figure]

Figure 1: Figure 1: Spatial autocorrelation of residuals

Ottar N. Bjornstad (2016). ncf: Spatial Nonparametric Covariance Functions. R package version 1.1-7. http://CRAN.R-project.org/package=ncf

Cliff, A. D., and J. K. Ord. (1981). Spatial Processes—Models and Applications. London: Pion.

Kühn, I. and Dormann, C. F.: Less than eight (and a half) misconceptions of spatial analysis, J. Biogeogr., 39(5), 995–998, 2012.

Xu, J., Morris, P. J., Liu, J. and Holden, J.: PEATMAP: Refining estimates of global peatland distribution based on a meta-analysis, Catena, 160(Supplement C), 134–140, 2018.

---

## Editor Comment (EC1) · G. A. Folberth (Editor) · 22 Dec 2017

**Reply to authors final response**

**Michel Bechtold, KU Leuven, December 20, 2017**

I thank the authors for their kind reply on my comment and for bringing up the interesting reference on "misconceptions of spatial analysis" by Kühn and Dormann (2012).

I am writing on response to that to clarify the issue. I do not consider spatial auto-correlation as a problem 'per se'. And I agree that peatland locations are very much spatially correlated in reality. A good model should thus also show this property and authors' check of the spatial autocorrelation of the residuals is a good idea. But it cannot be used as a proof for a well-calibrated model alone. My concerns were about the **validation approach.** The cross-validation followed a fully random design and must, however, be clustered or blocky, to guarantee that validation data is independent. There is a new paper published by Roberts et al. (2017 in Ecography) that in detail explains the issue. "Random hold-out data are too optimistic and favour overfitted models" (Carsten Dormann, personal communication, Dec 19, 2017). I think this is a very important point we must all be aware of because machine learning applications are spreading and often used with little care in that respect.

Sincerely,

Michel Bechtold

References:

Roberts, D. R., Bahn, V., Ciuti, S., Boyce, M. S., Elith, J., Guillera-Arroita, G., Hauenstein, S., Lahoz-Monfort, J. J., Schröder, B., Thuiller, W., Warton, D. I., Wintle, B. A., Hartig, F. and Dormann, C. F. (2017), Cross-validation strategies for data with temporal, spatial, hierarchical, or phylogenetic structure. Ecography, 40: 913–929. doi:10.1111/ecog.02881